# Retrospective analysis assessing the spatial and temporal distribution of paediatric acute respiratory tract infections in Ho Chi Minh City, Vietnam

Nhan Thi Ho,[1,2] Corinne Thompson,[1,3,4] Le Nguyen Thanh Nhan,[5] Hoang Minh Tu Van,[6] Nguyen Thanh Dung,[7] Phuc Tran My,[1] Vo Minh Quang,[7] Ngo Ngoc Quang Minh,[5] Tran Anh Tuan,[5] Nguyen Thanh Hung,[5] Ha Manh Tuan,[6] Nguyen Van Vinh Chau,[7] Marcel Wolbers,[1,3] Guy E Thwaites,[1,3] Marc Choisy,[8] Stephen Baker[1,3,9]

NTH and CT contributed equally.

For numbered affiliations see end of article.

**Correspondence to**
Dr Stephen Baker;
sbaker@oucru.org

## ABSTRACT

**Background** Acute respiratory tract infections (ARIs) are the leading cause of morbidity and mortality in young children in low/middle-income countries. Using routine hospital data, we aimed to examine the spatial distribution, temporal trends and climatic risk factors of paediatric ARIs in Vietnam.

**Methods** Data from hospitalised paediatric (<16 years) patients with ARIs residing in Ho Chi Minh City (HCMC) between 2005 and 2010 were retrieved from the two main Children's Hospitals and the Hospital for Tropical Diseases in HCMC. Spatial mapping and time series analysis were performed after disaggregating data into upper respiratory tract infections (URIs) and lower respiratory tract infections (LRIs).

**Results** Over the study period, there were 155 999 paediatric patients admitted with ARIs (33% of all hospital admissions). There were 68 120 URIs (14%) and 87 879 LRIs (19%). The most common diagnoses were acute pharyngitis (28% of all ARI), pneumonia (21%), bronchitis (18%) and bronchiolitis (16%). A significant increasing trend over time was found for both URIs (mean weekly incidence per 1000 population, I=3.12), incidence rate ratio for 1-week increase in time (RR 1.0, 95% CI 1.02 to 1.17) for URI and (I=4.02, RR 1.08 (95% CI 1.006 to 1.16)) for LRI. The weekly URI incidence peaked in May–June and was significantly associated with lags in weekly URI incidence and the average humidity, rainfall and water level. The weekly LRI incidence exhibited significant seasonality (P<0.0001), with an annual peak in September–October and was significantly associated with lags in weekly LRI incidence and lags in weekly average temperature, rainfall and water level.

**Conclusions** ARIs are a leading cause of childhood hospitalisation in HCMC, Vietnam. The incidence of ARIs was higher in the wet season and in specific HCMC districts. These results may guide health authorities in where and when to effectively allocate resources for the prevention and control of ARIs.

## Strengths and limitations of this study

► Using routine data from electronic hospital databases, we were able to retrieve (based on International Classification of Disease version 10 (ICD-10) codes) access information on paediatric admissions due to acute respiratory tract infections (ARIs) over a 6-year period in Ho Chi Minh City to examine the spatial distribution, temporal patterns and climatic risk factors of paediatric ARIs.

► Paediatric ARI admissions were mapped and compared with all other paediatric admissions to assess the spatial distribution of ARI admissions in comparison with the spatial referral pattern of the study hospitals.

► Data were stratified in two clinically differentiable groups: upper respiratory tract infections (URIs) and lower respiratory tract infections (LRIs) to use the general ICD-10 codes in the databases.

► Our findings were generated using hospitalised patient data and thus might not be generalisable for all outpatients. This may be particularly true for URIs, as many patients with URIs do not require hospitalisation.

► There was limited data on the specific pathogens associated with ARIs in the databases. URIs and LRIs are composed of various diseases associated with a wide range of pathogens which may have different spatiotemporal patterns.

## INTRODUCTION

Acute respiratory tract infections (ARIs) are composed of upper respiratory tract infections (URIs) and lower respiratory tract infections (LRIs). URIs are associated with infections at or above the larynx, are most commonly induced by viruses and include rhinitis (the common cold), sinusitis, ear

infections, acute pharyngitis, epiglottitis and laryngitis. LRIs are composed of bacterial or viral infections occurring below the larynx and encompass bronchitis, bronchiolitis and pneumonia.[1] ARIs are the leading cause of morbidity and mortality in young children,[2–4] accounting for >18% of all childhood deaths globally.[5] In 2010, there were approximately 120 million episodes and 1.3 million deaths associated with pneumonia worldwide, the vast majority of these occurred in Southeast Asia and sub-Saharan Africa.[5 6] In low-income and middle-income countries, ARIs in children aged <5 years are generally more severe than in older children and adults, resulting in a higher case-fatality than in high-income countries respectively.[1 7] This disparity in mortality rate is multifactorial and associated with differing aetiological agents, available therapies, exposure frequencies and host immunity.

In Ho Chi Minh City (HCMC), Vietnam, ARIs are among the most common cause of childhood hospitalisations.[8–10] The common identified pathogens of hospitalised ARI in HCMC include respiratory syncytial virus (RSV), rhinovirus, coronavirus, influenza and bocavirus,[8–10] in addition to the bacterial pathogens *Streptococcus pneumoniae* and *Haemophilus influenzae* type b.[11 12] With limited access to costly vaccines for bacterial aetiologies and a lack of vaccines for viral causes of ARIs, a greater understanding of the risk factors and subpopulations at risk of ARIs in HCMC is necessary to effectively focus existing prevention measures such as the encouragement of breast feeding, hand washing with soap and a reduction in household air pollution.[2]

There are data to suggest that the incidence of ARIs is associated with specific spatial, meteorological and sociodemographic risk factors.[13–16] However, specific temporal and climatic risk factors for both URIs and LRIs in HCMC have not been investigated. Here, we aimed to examine the spatial distribution, temporal patterns and climatic risk factors for hospitalised paediatric ARIs in HCMC. This analysis was performed to generate a better understanding of the factors that may influence ARIs and in order to guide healthcare resources for this common paediatric disease syndrome.

## METHODS
### Data sources
Data for this study were derived from the electronic hospital databases of all paediatric (<16 years) inpatient admissions to three large tertiary hospitals located in central HCMC: Children's Hospital 1 (CH1), Children's Hospital 2 (CH2) and the Hospital for Tropical Diseases (HTD). Data from the two Children's Hospitals were collected between 2005 and 2010 (inclusive) while the data from HTD were available from 2008 to 2010 only.

ARIs are caused by a range of differing pathogens and may have different spatiotemporal patterns and associations with climate and other covariates. Therefore, we allocated respiratory infections into two major clinically differentiable groups based on the International

Classification of Disease version 10 (ICD-10) codes version 2016 outlined by the WHO: (1) URIs were amalgamated under the ICD-10 codes J00:J06 and J9:J11 for influenza-like illnesses (ILI) and other acute URIs and (2) LRIs which incorporated the ICD-10 codes J12:J18 and J20:J22 for pneumonia and other acute LRIs. For the purposes of this study, the data from patients with the selected ICD-10 codes who resided in HCMC were extracted and analysed. Each patient record included data regarding age, sex, date of admission and discharge, ICD-10 code at discharge and residential address.

HCMC is a large economic centre in the South of Vietnam with a population of >8 million people. Geographically, it is divided into 24 districts (land areas ranging from 4.18 km$^2$ to 704.22 km$^2$) which are subdivided in 322 smaller administrative wards. There are five rural districts (population density ranging from 100 persons/km$^2$ to 3326 persons/km$^2$) and 19 urban districts (population density ranging from 2360 persons/km$^2$ to 45 582 persons/km$^2$). There are at least one local (district) hospital in each district, which serves both outpatients and inpatients of all ages. However, most local hospitals, especially those in urban districts receive mostly adult outpatients. There are two central hospitals (CH1 and CH2) that specialise in serving the paediatric population in HCMC. Another further central hospital (HTD) specialises in infectious diseases and also receives paediatric patients. While paediatric outpatients in HCMC may choose to visit local or private healthcare facilities or tertiary hospitals, most paediatric patients requiring hospitalisation in HCMC are referred or self-referred to one of the three study hospitals. Therefore, the three study hospitals (CH1, CH2 and HTD) receive the majority of paediatric inpatients from HCMC.

There are two main seasons in HCMC: a dry season from December to April and a rainy season from May to November. Population and district characteristic data (urban vs rural districts, district population density, district economic indicators (expenditure of district budget, revenues of district budget), district agricultural indicators (number of pigs in herds >2 months of age by district, percentage area of district used as rice paddy) and district education indicator (number of primary and secondary schools per district) were obtained from the HCMC Statistical Office.[17] As population figures for those aged <16 years per ward or district was not obtainable, the total population was used to estimate disease incidence. Patient addresses in HCMC were geocoded to districts and wards. City level weekly average climate data during 2005–2010 were obtained from the Ministry of Natural Resources and Environment of Vietnam including relative humidity (median=76.5%, range=61.7%–88%), temperature (median=28.1°C, range=24.4°C–31.8°C), rainfall (median=24.4 mm, range=0 mm–230.7 mm) and water level of the Dong Dien River (median=4.93 cm, range=−34.43 cm to 49.43 cm).[18]

## Spatial mapping

The variance of raw ward ARI incidence rates (number of ARI cases per 1000 ward total population for the whole study period) may largely vary due to the large variation in ward population size in HCMC. To take into account the spatial heterogeneity of variance and any dependence of ARI incidence rates between wards, ward ARI incidence rates were smoothed using local empirical Bayes estimates for rates reduced to a neighbourhood mean.[19] The neighbourhoods were selected using the neighbourhood list based on wards with contiguous boundaries (R package spdep V.0.5–88).[20] Kulldorff and Nagarwalla's method over the centroids was be used to scan for clusters of ARIs.[21] This method allows the detection of possible cluster of any size, at any location in a population with inhomogeneous spatial density of HCMC. A Poisson generalised linear model was used to explore the relationship between district ARI incidence and district characteristics. CIs and tests were based on robust SEs to control for mild violation of the distributional assumption that the variance equals the mean.[22]

## Time series analysis

City-wide ARI weekly incidences (number of cases per 1000 population) were examined for seasonality, time trends and associations with climatic covariates by Poisson generalised additive mixed models (GAMMs) implemented in the R package mgcv V.1.8–7.[23 24] The seasonal cycles and time trends of weekly ARI incidence were evaluated by the basic GAMM models containing the week of year (Ws=1–52) as cyclic cubic regression splines and the week of the whole study period (Wt=1–312) as cubic regression spline. As weekly ARI incidence may be associated with the ARI incidence and climatic condition of the previous weeks, in addition to the elements in the basic model, the full GAMM model contained lags of ARI weekly incidence up to 8 weeks, concurrent climatic condition and lags of climatic condition up to 8 weeks. The performance of the GAMM models was evaluated using adjusted $R^2$ and model diagnostic plots. Temporal correlation of the model residuals was evaluated by autocorrelation function and partial autocorrelation function.

All analyses and results were based on the records with available relevant information (complete cases). Spatial mapping and statistical analyses were stratified by URIs and LRIs and all analyses were performed in R V.3.2.2.[25]

## RESULTS

### Basic features of ARIs in the paediatric population of HCMC

From 2005 to 2010, there were 155 999 children (<16 years old) residing in HCMC admitted to the three study hospitals for ARIs (33% of all admissions); 323 245 patients were admitted for conditions other than ARIs. There were 68 120 patients with ICD-10 codes synonymous with acute URIs (14% of all admissions) and 87 879 patients with ICD-10 codes associated with acute LRIs (19% of all admissions). The proportion of men (62%) and women (38%) was comparable between URIs and LRIs. The median age of the patients was 1.5 years (IQR 0.7–2.9 years). However, patients with LRIs were significantly younger than those with URIs (median age in years (IQR)=1.1 (0.5–2.1) vs 2.1 (1.1–3.9)), Wilcoxon's test P<0.0001) and patients with LRI were hospitalised significantly longer than those with URIs (median days of hospitalisation (IQR)=6.0 (4–9) vs 4.0 (2–5), Wilcoxon's test P<0.0001) (table 1).

Overall, the most common diagnoses were acute pharyngitis (28% of all ARIs), pneumonia (21% of all ARIs), bronchitis (18% of all ARIs) and bronchiolitis (16% of all ARIs). Among the patients with URI, the majority of patients were diagnosed with acute pharyngitis (63% of all URIs), acute URIs of multiple unspecified sites (14%), acute tonsillitis (8%), acute laryngitis and tracheitis (6%), and acute nasopharyngitis (6%). Among the patients with LRI, the most common diagnoses were pneumonia with an unspecified organism (37% of all LRIs), acute bronchitis (32%), acute bronchiolitis (29%) and bacterial pneumonia not elsewhere classified (2%) (table 1).

### The spatial distribution of paediatric ARIs in HCMC

The raw ward-level incidence rates of ARIs for the entire study period ranged from 0 to 50 per 1000 total population (median (IQR)=18.8 (14.2–23.5)) and the smoothed ward-level empirical Bayesian estimated ARI incidence rates ranged from 1.2 to 48.8 ARI cases per 1000 population (median (IQR)=18.6 (14.7–23.7)). High incidence rates (>30 cases/1000 population) of ARI were observed in the wards surrounding the three study hospitals (wards in districts 1 and 4 in central HCMC) and in multiple wards located outside the central region of the city, specifically wards located in districts 2, 7 and Nha Be (observed vs expected incidence rate ratio (RR) >1.4, P<0.0001 using Kulldorff and Nagarwalla's scan) (figure 1A). Notably, the spatial distribution of ARIs and diseases other than ARI were comparable (figure 1A,B). We observed an increase with time in ward incidence of ARIs in the majority of the wards with the most apparent high incidences in wards in the south of the city (online supplementary figure S1). We additionally found the spatial distribution of URIs and LRIs were comparable (online supplementary figure S2).

The district level incidence over the entire study period of ARIs was significantly lower in the districts located further away from the hospitals (mean district incidence=26.7 per 1000 population, RR for two times increase in distance to hospital=0.88, 95% CI 0.80 to 0.96) and in rural districts (RR for urban vs rural districts=1.54, 95% CI 1.10 to 2.15) (figure 1A). Other district characteristics were not significantly associated with district ARI incidence.

### Temporal patterns and climatic risk factors of paediatric ARIs

The mean of weekly incidence of URIs for the study period was 3.12/1000 population. Weekly URI incidence showed a relatively small increase with time, with a difference in mean weekly URI incidence at the end and the

**Table 1** The characteristics and diagnoses for all patients with ARIs on admission

| Characteristic | All patients with ARI (n=155 999) | URI (n=68 120) | LRI (n=87 879) | P value |
|---|---|---|---|---|
| Male sex* | 96 930/155 997 (62%) | 42 251/68 119 (62%) | 54 679/87 878 (62%) | 0.43† |
| Age (year)‡ | 1.5 (0.7, 2.9) | 2.1 (1.1, 3.9) | 1.1 (0.5, 2.1) | <0.0001§ |
| Days in hospital¶ | 5.0 (3.0, 7.0) | 4.0 (2.0, 5.0) | 6.0 (4.0, 9.0) | <0.0001§ |
| ICD-10 code at discharge | | | | |
| Acute bronchiolitis (J21) | 25 123/155 999 (16%) | 0/68 120 (0%) | 25 123/87 879 (29%) | |
| Acute bronchitis (J20) | 28 202/155 999 (18%) | 0/68 120 (0%) | 28 202/87 879 (32%) | |
| Acute laryngitis and tracheitis (J04) | 4323/155 999 (3%) | 4323/68 120 (6%) | 0/87 879 (0%) | |
| Acute nasopharyngitis (J00) | 4049/155 999 (3%) | 4049/68 120 (6%) | 0/87 879 (0%) | |
| Acute obstructive laryngitis and epiglottitis (J05) | 9/155 999 (0%) | 9/68 120 (0%) | 0/87 879 (0%) | |
| Acute pharyngitis (J02) | 43 148/155 999 (28%) | 43 148/68 120 (63%) | 0/87 879 (0%) | |
| Acute sinusitis (J01) | 592/155 999 (0%) | 592/68 120 (1%) | 0/87 879 (0%) | |
| Acute tonsillitis (J03) | 5123/155 999 (3%) | 5123/68 120 (8%) | 0/87 879 (0%) | |
| Acute URIs of multiple and unspecified sites (J06) | 9300/155 999 (6%) | 9300/68 120 (14%) | 0/87 879 (0%) | |
| Bacterial pneumonia, not elsewhere classified (J15) | 2016/155 999 (1%) | 0/68 120 (0%) | 2016/87 879 (2%) | |
| Influenza due to other identified virus (J10) | 381/155 999 (0%) | 381/68 120 (1%) | 0/87 879 (0%) | |
| Influenza, virus not identified (J11) | 1195/155 999 (1%) | 1195/68 120 (2%) | 0/87 879 (0%) | |
| Pneumonia due to *Haemophilus influenza* (J14) | 11/155 999 (0%) | 0/68 120 (0%) | 11/87 879 (0%) | |
| Pneumonia due to other infectious organisms, not elsewhere classified (J16) | 26/155 999 (0%) | 0/68 120 (0%) | 26/87 879 (0%) | |
| Pneumonia due to *Streptococcus pneumonia* (J13) | 16/155 999 (0%) | 0/68 120 (0%) | 16/87 879 (0%) | |
| Pneumonia in diseases classified elsewhere (J17) | 1/155 999 (0%) | 0/68 120 (0%) | 1/87 879 (0%) | |
| Pneumonia, organism unspecified (J18) | 32 371/155 999 (21%) | 0/68 120 (0%) | 32 371/87 879 (37%) | |
| Unspecified acute LRI (J22) | 7/155 999 (0%) | 0/68 120 (0%) | 7/87 879 (0%) | |
| Viral pneumonia, not elsewhere classified (J12) | 106/155 999 (0%) | 0/6820 (0%) | 106/87 879 (0%) | |

*Data from 155 997 patients.
†$\chi^2$ test comparing URIs versus LRIs.
‡Data from 147 420 patients.
§From Wilcoxon's rank-sum test comparing URIs versus LRIs.
¶Data from 96 362 patients.
ARI, acute respiratory tract infection; ICD-10, International Classification of Disease version 10 (ICD-10) codes; LRIs, lower respiratory tract infections; URIs, upper respiratory tract infections.

start of the study period of 0.86 patients per 1000 population. The basic Poisson GAMM model indicated that URI admissions in the study hospitals exhibited a significant increasing trend with time (P<0.0013) and seasonality (P<0.0001) with a peak in May–June and a smaller peak in December–January (figure 2A).

The mean weekly LRI incidence for the study period was 4.02/1000 population. There was a significant increase of weekly LRI incidence over time (P=0.0037) with a substantial difference in the mean weekly LRI incidence at the end versus the start of the study period of 1.54 patients/1000 population. The seasonal cycle of LRIs (P<0.0001) was more obvious than that of URIs, with a single, wide annual peak in the months of September and October (figure 2B).

In a univariate analysis with Poisson generalised linear model, we found that weekly URI incidence was significantly associated with weekly average temperature (RR for 1° increase of temperature=1.048, 95% CI 1.030 to 1.066) and humidity (RR for 1% increase in relative

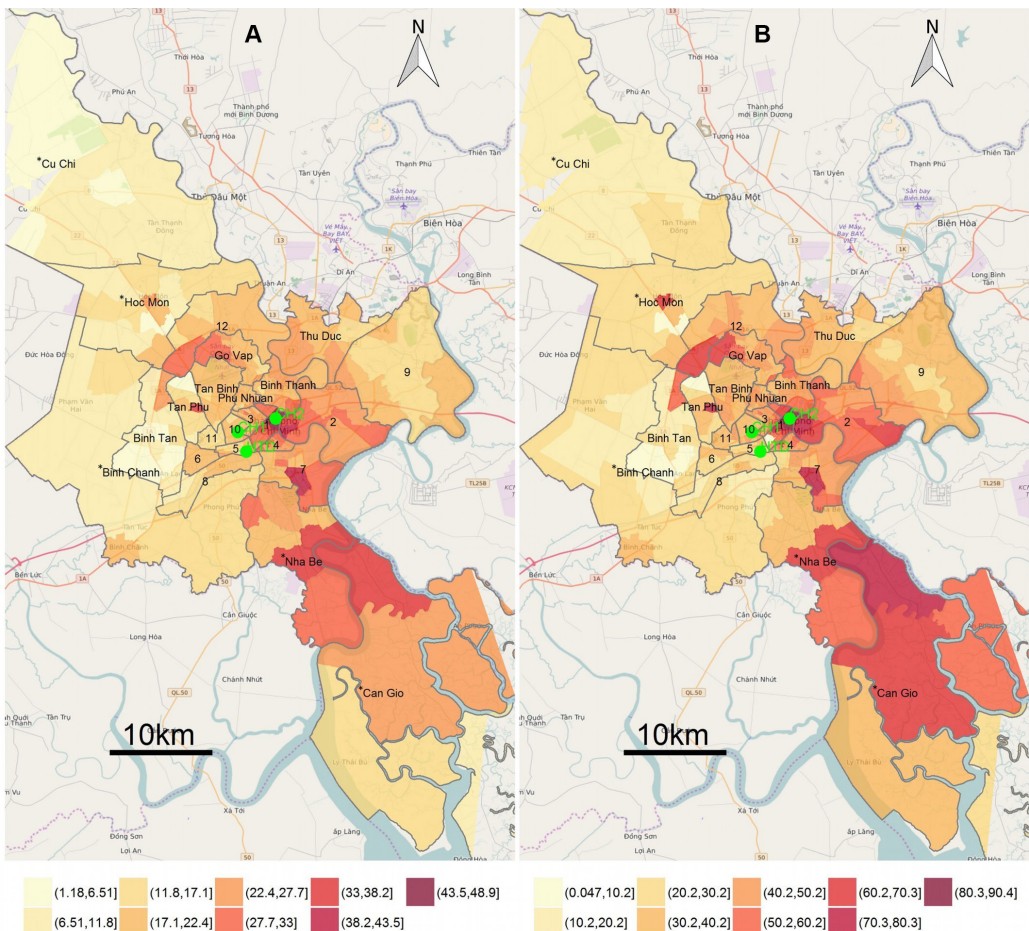

**Figure 1** Maps for empirical Bayesian estimated incidence rates of acute respiratory tract infections (ARIs) and other diseases at ward level: (A) ward-level empirical Bayesian estimated incidence rates (EBR) per 1000 population for ARIs with locations of the three study hospitals in green, (B) ward-level EBR for other diseases rather than ARIs. District names and district boundaries (grey) are shown. *Rural districts. CH1, Children Hospital 1; CH2, Children Hospital 2; HTD, Hospital for Tropical Diseases.

humidity=1.006, 95% CI 1.001 to 1.011). Further, weekly LRI incidence was significantly associated with weekly average humidity (RR=1.030, 95% CI 1.024 to 1.036) and rainfall (RR for 1 mm increase in rainfall=1.019, 95% CI 1.014 to 1.024) (figure 3). However, after controlling for seasonality, time trend, lags of weekly incidence and other climatic covariates in the full GAMM model, the weekly URI incidence was significantly associated with 1-week and 2-week lags in URI incidence (RR=1.202, 95% CI 1.159 to 1.245 and RR=1.063, 95% CI 1.019 to 1.109, respectively), 1-week and 3-week lags in weekly average humidity (RR=1.009, 95% CI 1.003 to 1.014 and RR=0.990, 95% CI 0.984 to 0.996, respectively), a 3-week lag in weekly average rainfall (RR=1.006, 95% CI 1.003 to 1.009) and a 4-week lag in weekly average river water level (RR for 1 cm increase in water level=1.003, 95% CI 1.002 to 1.005) (table 2).

After adjusting for other factors in the full GAMM model, weekly LRI incidence was significantly associated with a 1-week and 2-week lags in LRI incidence (RR=1.173, 95% CI 1.138 to 1.208 and RR=1.053, 95% CI 1.015 to 1.092, respectively), a 1-week lag in weekly average temperature (RR=0.976, 95% CI 0.953 to 0.998),

a 1-week lag in weekly average rainfall (RR=1.004, 95% CI 1.001 to 1.007) and a 1-week lag in weekly average water level (RR=0.998, 95% CI 0.997 to 1.000) (table 2). The time trends remained significant for both URIs (incidence RR for 1-week increase in time (RR)=1.09, 95% CI 1.02 to 1.17, P=0.0096) and LRIs (RR=1.08, 95% CI 1.006 to 1.16, P=0.0345) while only LRIs exhibited significant seasonality (P<0.0001) in the full GAMM models. The full GAMM models fitted the data well for both URIs (adjusted R$^2$=0.72) and LRIs (adjusted R$^2$=0.88) (figure 2 and online supplementary figure S3). Temporal correlation of weekly incidence was well controlled for URIs and also relatively well controlled for LRIs in the full GAMM models (online supplementary figure S4).

## DISCUSSION

Our study examined the spatial distribution, temporal patterns and climatic risk factors of ARIs in HCMC, Vietnam. The findings were based on the data of a large number of ARI inpatients retrieved from routinely collected electronic hospital databases over a 6-year period. Our data showed that ARIs accounted for

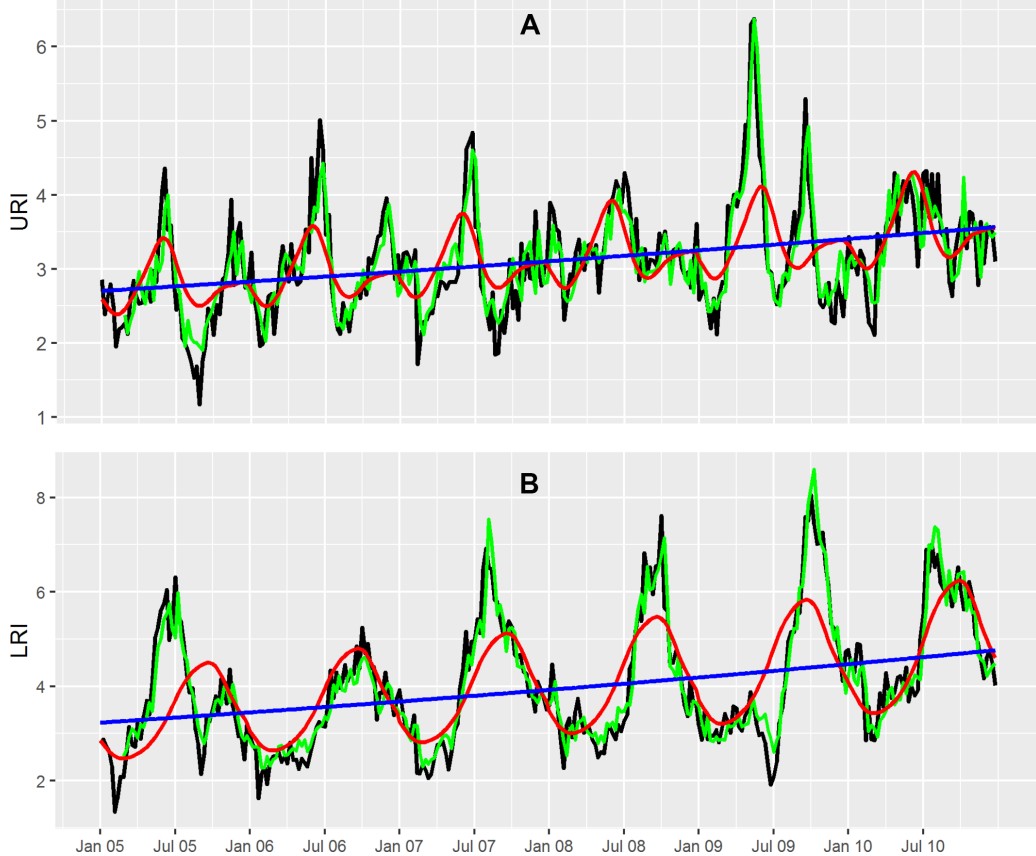

**Figure 2** Plots of the city-wide weekly incidence time series for upper respiratory tract infections (URIs) and lower respiratory tract infections (LRIs) from the generalised additive mixed model (GAMM). (A) Smoothed seasonal cycles (red line), time trend (blue line) from the basic Poisson GAMM model, fitted values (green line) from the full Poisson GAMM model and observed values (black line) of city-wide weekly incidence (number of cases per 1000 population) of URIs. (B) Smoothed seasonal cycles (red line), time trend (blue line) from the basic Poisson GAMM model and fitted values (green line) from the full Poisson GAMM model and observed values (black line) of city-wide weekly incidence of LRIs.

one-third of all hospital admissions over the study period and were the most common cause of hospitalised diseases in children in HCMC. The spatial distribution of ARIs was comparable to the spatial referral patterns of study hospitals in HCMC (the spatial distribution of diseases other than ARIs). High incidence of ARIs was found not only in districts close to the study hospitals (districts 1 and 4) but also in districts 2, 7 and Nha Be, which are located further away from three study hospitals. This finding is likely a result of urbanisation or better economic status of those living in these districts.

The observed increase in hospitalised ARI incidence with time may be explained by better healthcare awareness within the population, through better access to tertiary hospitals, better healthcare policies (free healthcare for children aged <6 years was initiated in Vietnam) or a real increase in ARI incidence. The city-wide increase of ARI incidence, especially LRIs, may also be attributed to an increase in air pollution. Pollution was shown to be associated with an increase in LRI incidence in study conducted by the Collaborative Working Group on Air Pollution, Poverty and Health in HCMC.[26] The number of beds in these tertiary hospitals did not increase over the study period, and the policy of these hospitals is to

accommodate all patients visiting the hospitals regardless of the capacity of the hospital and staff. An increase in ARI admission indicates an increase in burden for these tertiary hospitals and an increased workload for healthcare, which may impact negatively on the quality of care. Therefore, there should be effective policies to improve the allocation of healthcare resources in these hospitals as well as to improve the quality and capacity of local healthcare facilities to share this increasing disease burden.

The weekly incidence of both URIs and LRIs showed significant association with 1-week and 2-week lags of incidence indicating that temporal correlation might play an important role in URI and LRI incidence. Furthermore, we additionally noted a difference in seasonal patterns of URIs (two small peaks a year) and LRIs (one strong peak a year) and a difference in association patterns with climatic factors of URIs and LRIs. These findings suggested that climatic factors might differentially influence the pathogens causing URIs and LRIs in HCMC. The association of URI incidence with 1-week and 3-week lags of weekly average humidity was consistent with the association between the seasonality of absolute humidity and ILI in Vietnam as previously reported.[27] In addition, the national surveillance of influenza and ILI in Vietnam

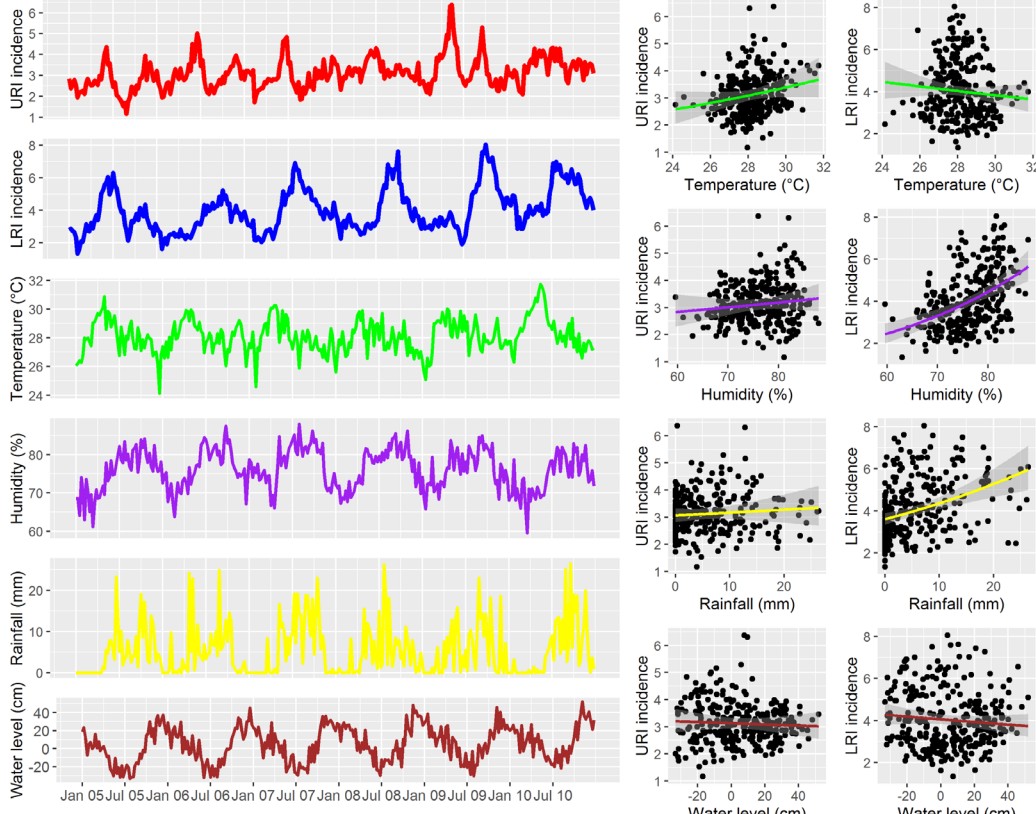

**Figure 3** Time series of city-wide weekly incidence of upper respiratory tract infections (URIs) and lower respiratory tract infections (LRIs) versus weekly climatic covariates. Left panel: time series of weekly URI and LRI incidence and weekly average climatic covariates. Right panel: scatter plots with fitted lines and 95% CI from univariate Poisson generalised linear model for weekly URI or LRI incidence versus each climatic covariate.

during the study period exhibited a similar seasonal pattern with two peaks a year at approximately the same months with the two peaks of URIs in our data.[28] The exceptional high peak of URIs without a concurrent peak of LRIs in 2009 also occurred simultaneously with 2009 H1N1. Therefore, influenza and ILI were likely key contributors to the seasonality and the peaks of URIs in our data. Moreover, a study conducted at CH2 investigating viral ARI pathogens found that the peak of RSV infections occurred in rainy season (May–November in HCMC). This period included the peak of LRI months and the larger URI peak.[10] Furthermore, RSV is also thought to predispose infected hosts to bacterial infections.[29] In addition, a sharp increase of LRI incidence was observed soon after the peak of URIs (May–June) suggests that URIs and LRIs share some common epidemiological features. As a consequence, appropriate treatment and transmission control for URIs may impact on the incidence of later hospitalisation due to LRIs. As the peaks of hospitalisation for LRIs are in September–October, resources for paediatric healthcare should be prepared and local healthcare facilities should be enhanced to share the increase in hospitalisation caused by LRIs at tertiary hospitals during this period.

A large proportion of ARI admissions in these data were URIs, two-third of these were diagnosed with acute pharyngitis. These conditions can usually be treated in outpatient settings and do not typically require hospitalisation. However, the decision to hospitalise patients with ARI in these tertiary hospitals is not strictly based on WHO Integrated Management of Childhood Illness (IMCI) guidelines and is also dependent on the preferences of caregivers and the subjective patient evaluation of the attending physicians. Therefore, there should stricter policies for applying IMCI guidelines and other clinical criteria to reduce unnecessary URI admissions in these hospitals, as well as to improving outpatient care and follow-up.

Our study has limitations as the findings were generated using routine data of from hospitalised patients. Therefore, are finding may be generalisable for outpatients, especially for URIs, as many patients with URI may not be admitted to hospital. The findings for LRIs may be more generalisable as LRIs are more severe than URIs and more likely to be hospitalised. Second, the databases contained mostly general ICD-10 codes and very limited specific diagnostic information regarding the ARI pathogens. Different pathogens can cause differing URI and LRI disease presentations and different pathogens may be influenced differently by climatic, environmental, population factors as well as host immunity. Therefore, further surveillance and studies investigating different groups of pathogens for ARIs are necessary.

**Table 2** Results of the full generalised additive mixed models (GAMMs) for upper respiratory tract infections (URIs) and lower respiratory tract infections (LRIs)

| | URIs | | | | LRIs | | | |
|---|---|---|---|---|---|---|---|---|
| | RR | LL | UL | P value | RR | LL | UL | P value |
| Intercept | 0.340 | 0.133 | 0.869 | 0.0251 | 1.421 | 0.381 | 5.303 | 0.6013 |
| Lag 1-week incidence | 1.202 | 1.159 | 1.245 | 0.0000 | 1.173 | 1.138 | 1.208 | 0.0000 |
| Lag 2-week incidence | 1.063 | 1.019 | 1.109 | 0.0048 | 1.053 | 1.015 | 1.092 | 0.0060 |
| Lag 3-week incidence | 0.999 | 0.957 | 1.042 | 0.9470 | 0.987 | 0.951 | 1.024 | 0.4861 |
| Lag 4-week incidence | 0.974 | 0.934 | 1.017 | 0.2328 | 0.998 | 0.962 | 1.035 | 0.9180 |
| Temperature (°C) | 1.017 | 0.996 | 1.040 | 0.1178 | 1.006 | 0.985 | 1.028 | 0.5748 |
| Lag 1-week temperature | 0.990 | 0.966 | 1.015 | 0.4480 | 0.976 | 0.953 | 0.998 | 0.0348 |
| Lag 2-week temperature | 1.019 | 0.994 | 1.045 | 0.1389 | 1.007 | 0.984 | 1.030 | 0.5620 |
| Lag 3-week temperature | 1.010 | 0.985 | 1.035 | 0.4470 | 1.019 | 0.995 | 1.043 | 0.1186 |
| Lag 4-week temperature | 1.012 | 0.990 | 1.035 | 0.2709 | 1.015 | 0.993 | 1.038 | 0.1722 |
| Humidity (%) | 0.998 | 0.993 | 1.004 | 0.5619 | 0.998 | 0.993 | 1.004 | 0.5506 |
| Lag 1-week humidity | 1.009 | 1.003 | 1.014 | 0.0051 | 0.997 | 0.991 | 1.002 | 0.2716 |
| Lag 2-week humidity | 1.002 | 0.996 | 1.007 | 0.6000 | 1.001 | 0.995 | 1.006 | 0.7603 |
| Lag 3-week humidity | 0.990 | 0.984 | 0.996 | 0.0008 | 1.000 | 0.995 | 1.006 | 0.9237 |
| Lag 4-week humidity | 1.005 | 0.999 | 1.010 | 0.1024 | 1.000 | 0.994 | 1.005 | 0.8610 |
| Rainfall (mm) | 1.000 | 0.997 | 1.003 | 0.9670 | 1.001 | 0.998 | 1.004 | 0.5615 |
| Lag 1-week rainfall | 1.001 | 0.997 | 1.004 | 0.7145 | 1.004 | 1.001 | 1.007 | 0.0073 |
| Lag 2-week rainfall | 1.002 | 0.998 | 1.005 | 0.2979 | 1.000 | 0.998 | 1.003 | 0.7548 |
| Lag 3-week rainfall | 1.006 | 1.003 | 1.009 | 0.0005 | 1.001 | 0.998 | 1.004 | 0.4927 |
| Lag 4-week rainfall | 0.997 | 0.994 | 1.001 | 0.1163 | 0.999 | 0.997 | 1.002 | 0.6949 |
| Water level (cm) | 0.999 | 0.997 | 1.000 | 0.1519 | 0.999 | 0.997 | 1.000 | 0.0751 |
| Lag 1-week water level | 1.001 | 0.999 | 1.002 | 0.2120 | 0.998 | 0.997 | 1.000 | 0.0111 |
| Lag 2-week water level | 0.999 | 0.998 | 1.001 | 0.2488 | 1.000 | 0.998 | 1.001 | 0.7821 |
| Lag 3-week water level | 1.000 | 0.998 | 1.001 | 0.7223 | 1.000 | 0.999 | 1.002 | 0.9576 |
| Lag 4-week water level | 1.003 | 1.002 | 1.005 | 0.0000 | 1.001 | 1.000 | 1.003 | 0.1473 |
| Time trend (week) | 1.092 | 1.022 | 1.166 | 0.0096 | 1.080 | 1.006 | 1.159 | 0.0345 |

Results of fixed effects (with only the results of lags up to 4 weeks) are shown.
P values for seasonal cycles from the full GAMM of URIs=0.5177 and LRIs <0.0001, respectively.
Adjusted $R^2$ of the full GAMM for URIs=0.72 and LRIs=0.88, respectively.
LL, lower limit; RR, rate ratio; UL, upper limit of 95% CI.

## CONCLUSION

We found that ARIs accounted for one-third of total hospital admission and were the leading cause of hospitalisation in children in HCMC, Vietnam. The spatial distribution of hospitalised ARIs was similar to the spatial referral patterns of the study hospitals with high incidences in the districts surrounding the locations of the three study hospitals and also in three districts further out the city. The seasonality of ARIs may be explained by the association of ARI incidence with various climatic factors (temperature, humidity and rainfall) and the seasonality of influenza, ILI and RSV. The increase of LRI incidence soon after the peak of URIs suggests that URIs and LRIs share common epidemiological factors or LRIs are complications of URIs. Understanding the spatial distribution, temporal patterns and climatic risk factors of ARIs should aid the health authorities in outbreak preparation and assist in allocating resources for the prevention and control of the diseases, especially in settings where vaccines for viral and bacterial pathogens of ARIs are not routinely available.

**Author affiliations**
[1]Oxford University Clinical Research Unit, Wellcome Trust Major Overseas Program, Ho Chi Minh City, Vietnam
[2]Pediatrics, Columbia UniversityMedical Center, New York, New York, United States
[3]Nuffield Department of Medicine, Centre for Tropical Medicine and Global Health, University of Oxford, Oxford, UK
[4]Infection Biology, The London School of Hygiene & Tropical Medicine, London, UK
[5]General medicine, Children's Hospital 1, Ho Chi Minh City, Vietnam
[6]General medicine, Children's Hospital 2, Ho Chi Minh City, Vietnam
[7]General planning, Hospital for Tropical Diseases, Ho Chi Minh City, Vietnam
[8]Institute of Research and Development, Ho Chi Minh City, Vietnam
[9]Department of Medicine, University of Cambridge, Cambridge, UK

**Acknowledgements** The authors are grateful to the staffs of Children's Hospital 1, Children's Hospital 2 and The Hospital for Tropical Diseases for providing the raw databases used in this study.

**Contributors** This study was conceptualised by NTHo, CT, MC and SB. Data were obtained by NTHo, CT, LNTN, HMTV, NTD, PTM, VMQ, NNQM, TAT, NTHu, HMT and NVVC. Data were analysed by NTHo, CT, MW and MC. NTHo, CT and SB wrote the manuscript with input from all authors. NTHo, CT, LNTN, HMTV, NTD, PTM, VMQ, NNQM, TAT, NTHu, HMT, NVVC, MW, GET, MC and SB read and approved the final version of the manuscript.

**Funding** This work was supported by the Wellcome Trust Vizions strategic award (WT1093824). SB is a Sir Henry Dale Fellow, jointly funded by the Wellcome Trust and the Royal Society (100087/Z/12/Z). NTH is currently funded by Mervyn Susser's fellowship at Columbia University Medical Center, USA.

**Competing interests** None declared.

**Patient consent** Not required.

**Ethics approval** Ethical approval was granted by all of the three study hospitals: Children Hospital 1, Children Hospital 2 and the Hospital for Tropical Diseases.

**Provenance and peer review** Not commissioned; externally peer reviewed.

**Data sharing statement** Data will be available on reasonable request to corresponding author.

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
