## [Reviewer comments · BMJ Open]

ARTICLE DETAILS

TITLE (PROVISIONAL)	A retrospective analysis assessing the spatial and temporal distribution of pediatric acute respiratory tract infections in Ho Chi Minh City, Vietnam
AUTHORS	Ho, Nhan Thi; Thompson, Corinne; Nhan, Le Nguyen Thanh; Van, Hoang Minh Tu; Dung, Nguyen Thanh; Tran My, Phuc; Quang, Vo Minh; Minh, Ngo Ngoc Quang; Tuan, Tran Anh; Hung, Nguyen Thanh; Tuan, Ha Manh; Vinh Chau, Nguyen Van; Wolbers, Marcel; Thwaites, Guy; Choisy, Marc; Baker, Stephen

VERSION 1 – REVIEW

REVIEWER	Hitoshi Oshitani Tohoku University Graduate School of Medicine, Japan
REVIEW RETURNED	19-Apr-2017

GENERAL COMMENTS	The manuscript describes the spatial and temporal patterns of pediatric cases admitted to three tertiary hospitals in Ho Chi Minh City, Vietnam analyzed by using the electronic hospital database. The manuscript is well written and provide some important insights for seasonality and geographic distribution of acute respiratory infections. However, I have several comments for the manuscript. 1) They did not describe general health care systems in Ho Chi Minh City. In rural areas, probably more patients go to local hospitals such as district hospitals. On the other hand, in urban areas, some patients may go to private hospitals. Brief description of general health care system in the city might be useful to understand the spatial distribution of cases. 2) The numbers of URI and LRI cases increased over time (Figure 2). The authors indicated better healthcare awareness, better access, and real increase of cases as possible reasons for such increase. However, hospital capacities in these three hospitals including bed capacity and number of healthcare staff might have increased during the study period. 3) They discusses RSV as an important contributor for seasonal pattern. Another obvious possible contributor is influenza viruses, which usually show some seasonality even in tropical area like Ho Chi Minh City. I understand influenza surveillance is being conducted in Ho Chi Minh City and the data on seasonality of influenza viruses should be available. They also should include some discussion on influenza viruses as a possible contributor for seasonal pattern especially for URI. Moreover, the study period included 2009 when pandemic (H1N1) 2009 occurred. Significantly higher peak of URI in mid-2009 without any clear peak of LRI might be due to pandemic (H1N1) 2009. This point also should be discussed. 4) In low and middle-income countries, most of pediatric admissions
--

	are due to LRI, particularly pneumonia because IMCI guidelines are widely used to determine admission criteria. However, in this study, the large proportion of cases were patients with URI. Probably less severe cases are also admitted in Vietnam. I think some discussion would be useful to understand the situation in Vietnam.
--	--

REVIEWER	Oyelola Adegboye Qatar University, Doha, Qatar
REVIEW RETURNED	17-Oct-2017

GENERAL COMMENTS	A retrospective analysis assessing the spatial and temporal distribution of pediatric acute respiratory tract infections in Ho Chi Minh City, Vietnam This manuscript is well written and focused on an important issue regarding spatio-temporal analysis of pediatric infections in a developing country. I have the following minor comments:  1. L170-175, provide brief conceptual framework for local empirical Bayes estimates and cluster scan. 2. L170-175, provide reference for local empirical Bayes estimates 3. L177, what is CNSI? It has not been used previously. Why is this mentioned here? 4. Mostly throughout the results section the reader is left wondering which table the authors interpreting, Table 1 or table 2.
---

VERSION 1 – AUTHOR RESPONSE

Reviewer: 1

The manuscript describes the spatial and temporal patterns of pediatric cases admitted to three tertiary hospitals in Ho Chi Minh City, Vietnam analyzed by using the electronic hospital database. The manuscript is well written and provide some important insights for seasonality and geographic distribution of acute respiratory infections. However, I have several comments for the manuscript.

1) They did not describe general health care systems in Ho Chi Minh City. In rural areas, probably more patients go to local hospitals such as district hospitals. On the other hand, in urban areas, some patients may go to private hospitals. Brief description of general health care system in the city might be useful to understand the spatial distribution of cases.

Response: We have added a brief description of general healthcare systems in Ho Chi Minh City to the Method sub-section 'Data source', page 7, line 157-166. In addition, we mapped ARIs in comparison with all pediatric hospital admissions due to other conditions rather than ARIs to evaluate the spatial distribution of ARIs in comparison with spatial referral patterns to the three study hospitals. We also revised the first paragraph of the "Discussion" section, page 12, line 291-297 to better address this.

2) The numbers of URI and LRI cases increased over time (Figure 2). The authors indicated better healthcare awareness, better access, and real increase of cases as possible reasons for such increase. However, hospital capacities in these three hospitals including bed capacity and number of healthcare staff might have increased during the study period.

Response: The number of beds in these tertiary hospitals did not increase during the study period. We did not identify any the documented facts about the increase of health care staff during the study period in the three study hospitals but if there was any increase in the number of staffs, the increase would be very little. By policy, these hospitals must accommodate all patients visited the hospitals and needed admission regardless of the capacity of the hospitals or the availability of their staffs. We revised the "Discussion" section, page 13, line 310-322 to better discuss this point.

3) They discusses RSV as an important contributor for seasonal pattern. Another obvious possible contributor is influenza viruses, which usually show some seasonality even in tropical area like Ho Chi Minh City. I understand influenza surveillance is being conducted in Ho Chi Minh City and the data on seasonality of influenza viruses should be available. They also should include some discussion on influenza viruses as a possible contributor for seasonal pattern especially for URI. Moreover, the study period included 2009 when pandemic (H1N1) 2009 occurred. Significantly higher peak of URI in mid-2009 without any clear peak of LRI might be due to pandemic (H1N1) 2009. This point also should be discussed.

Response: We admit that we missed this point in our previous version of the manuscript. We agree that influenza and influenza like illness might play an important role in the temporal patterns of URIs in our data. The reviewer was very right that there is surveillance for influenza and there are published influenza data in Ho Chi Minh City and Vietnam. We revised our "Discussion" section, page 13-14, line 324-348 to address this.

4) In low and middle-income countries, most of pediatric admissions are due to LRI, particularly pneumonia because IMCI guidelines are widely used to determine admission criteria. However, in this study, the large proportion of cases were patients with URI. Probably less severe cases are also admitted in Vietnam. I think some discussion would be useful to understand the situation in Vietnam.

Response: The reviewer is correct that most URIs do not require hospitalization and that IMCI have been used in Vietnam before, during and after the study period until now. However, in reality, in the three study hospitals, the decision to hospitalize ARI patients was not strictly based on IMCI guidelines and also depended on the preference of the patients' caregivers and the subjective patient evaluation of the attending physicians. Therefore, URIs contributed quite a large proportion (44%) of all ARI admissions in our data and many of those URI admissions were probably unnecessary. We added a paragraph to our "Discussion", page 14, line 350-358 to address this point.

Reviewer: 2

This manuscript is well written and focused on an important issue regarding spatio-temporal analysis of pediatric infections in a developing country.

I have the following minor comments:

L170-175, provide brief conceptual framework for local empirical Bayes estimates and cluster scan.

Response: We agree that it is useful to provide some conceptual framework for local empirical Bayes estimates and cluster scan. We revised the method sub-section 'Spatial mapping', page 8, line 182-190 to include this information.

Comment: L170-175, provide reference for local empirical Bayes estimates

Response: We added the reference for local empirical Bayes estimates (page 8, line 186)

Comment: L177, what is CNSI? It has not been used previously. Why is this mentioned here?

Response: Thank you for picking up this typo error. It should be “ARI”, not “CNSI” and we have corrected this (page 8, line 192).

Comment: Mostly throughout the results section the reader is left wondering which table the authors interpreting, Table 1 or table 2.

Response: Thank you so much for the comment. We added the annotation (e.g. “Table 1” or “Table 2”) to the results section to make it clearer which table was interpreted.